Improving insect conservation across heterogeneous landscapes using species–habitat networks

http://orcid.org/0000-0002-6726-1323 Cappellari Andree andree.cappellari@phd.unipd.it
http://orcid.org/0000-0001-7429-7685 Marini Lorenzo
Department of Agronomy, Food, Natural Resources, Animals and Environment (DAFNAE), University of Padua , Legnaro, Padua , Italy
Clough Yann
Electronic publication date: 2021 Jan 5
Publication date: 2021
Volume: 9
Electronic Location ID: e10563
Received 2020 Aug 25; Accepted 2020 Nov 22
Copyright: © 2021 Cappellari and Marini
Copyright year: 2021
Copyright holder: Cappellari and Marini
License: This is an open access article distributed under the terms of the Creative Commons Attribution License, which permits unrestricted use, distribution, reproduction and adaptation in any medium and for any purpose provided that it is properly attributed. For attribution, the original author(s), title, publication source (PeerJ) and either DOI or URL of the article must be cited.
License URL: https://creativecommons.org/licenses/by/4.0/

Keywords: Butterflies, Calcareous grasslands, Centrality, Community detection, Meadows, Network analysis

Funding: Papilionoidea e Hesperioidea SIC-IT 3310009 and SPA-IT 33110011 University of Padua The research was part of the project “Studio della biodiversità e delle relazioni tra i Lepidotteri Ropaloceri (Papilionoidea e Hesperioidea) e la vegetazione e dell’impatto delle principali pressioni antropiche” (management plan of the SIC-IT 3310009 “Magredi del Cellina” and the SPA-IT 33110011 “Magredi di Pordenone”), fully funded by the Friuli-Venezia Giulia region to Lorenzo Marini and Paolo Paolucci (University of Padua). The funders had no role in study design, data collection and analysis, decision to publish, or preparation of the manuscript.

==============================
Background

One of the biggest challenges in conservation is to manage multiple habitats for the effective conservation of multiple species, especially when the focal species are mobile and use multiple resources across heterogeneous protected areas. The application of ecological network tools and the analysis of the resulting species–habitat networks can help to describe such complex spatial associations and improve the conservation of species at the landscape scale.

Methods

To exemplify the application of species–habitat networks, we present a case study on butterflies inhabiting multiple grassland types across a heterogeneous protected area in North-East Italy. We sampled adult butterflies in 44 sites, each belonging to one of the five major habitat types in the protected area, that is, disturbed grasslands, continuous grasslands, evolved grasslands, hay meadows and wet meadows. First, we applied traditional diversity analyses to explore butterfly species richness and evenness. Second, we built and analyzed both the unipartite network, linking habitat patches via shared species, and the bipartite network, linking species to individual habitat patches.

Aims

(i) To describe the emerging properties (connectance, modularity, nestedness, and robustness) of the species–habitat network at the scale of the whole protected area, and (ii) to identify the key habitats patches for butterfly conservation across the protected area, that is, those supporting the highest number of species and those with unique species assemblages (e.g., hosting specialist species).

Results

The species–habitat network appeared to have a weak modular structure, meaning that the main habitat types tended to host different species assemblages. However, the habitats also shared a large proportion of species that were able to visit multiple habitats and use resources across the whole study area. Even butterfly species typically considered as habitat specialists were actually observed across multiple habitat patches, suggesting that protecting them only within their focal habitat might be ineffective. Our species–habitat network approach helped identifying both central habitat patches that were able to support the highest number of species, and habitat patches that supported rare specialist species.

Introduction

Protected areas play a fundamental role for the conservation of biodiversity. In many cases, however, these areas are composed of a mosaics of small patches of different habitat types, both managed and unmanaged ones. The conservation of insect diversity across such heterogeneous landscapes may face various problems, in particular when the focal species are mobile and use multiple resources across different habitat patches (Kremen et al., 2007; Marini et al., 2019). For instance, when landscapes are composed of small patches with a large perimeter-to-area ratio, the local communities are heavily impacted by the surrounding landscape (Krauss, Steffan-Dewenter & Tscharntke, 2003). Most of the decisions on how to manage single habitats for conservation are usually based on the results of diversity analyses where patches, habitats or interventions are usually ranked according to the number of species and individuals they support (see for example, Villemey et al., 2015; Ernst, Tscharntke & Batáry, 2017; Denning & Foster, 2018). While this approach can help identifying ideal local habitat quality to maximize species diversity, it also overlooks the potential interactions between multiple habitat patches in supporting communities of mobile organisms (Harlio et al., 2019). There have been several attempts to implement landscape-scale approaches to conservation encouraging bigger and larger number of protected areas, enhancing connectivity, and improving habitat quality (Albert et al., 2017; Donaldson, Wilson & Maclean, 2017), but little emphasis has been placed to develop tools to optimize conservation actions within protected areas characterized by large landscape heterogeneity at small spatial scale.

Managing multiple habitats for the conservation of multiple species can be challenging. Recently, it has been proposed to adapt network tools to describe such complex spatial interactions (Hackett et al., 2019; Marini et al., 2019) and to use the resulting species–habitat networks and their metrics to improve conservation of species at the landscape scale (Nardi et al., 2019; Pompozzi et al., 2019; Saunders & Rader, 2019; Lami et al., 2020). First, topology metrics can inform on the architecture and the emerging properties of the whole species–habitat network. On the one hand, in a protected area with a nested structure species-rich patches host both common and rare species, while species-poor patches are mainly visited by generalist species and so their loss is unlikely to have ripple effects on the entire protected area (Table 1A) (Patterson, 1987). On the other hand, in a protected area with a strong modular structure some species interact more frequently with some habitat types forming modules, so patches belonging to the same module are more tightly connected to each other than to patches belonging to different habitat types (Table 1B). In this scenario, different modules need to be considered as individual management blocks. Second, node-level metrics can describe properties of single habitat patches within the network. For instance, patch centrality can inform about the importance of single habitats and patches in supporting species across the whole protected area. A patch with high centrality hosts many species that also occur in other habitats, playing a fundamental role in supporting generalist species across the whole species–habitat network (Table 1C).

Table 1 Explanation and example of conservation implications of species–habitat network metrics, both at network and node level.

Metric	Explanation	Example of conservation implications	
Network level: species–habitat network architecture	
(A) Nestedness
	Species-rich patches host both common and rare species, while species-poor patches are only visited by generalist species.	A nested structure (high nestedness value) provides robustness against the loss of species-poor habitats, and the management should therefore focus on species-rich sites. A non-nested structure (low nestedness value) indicates a low level of robustness, so the management should focus on species-poor sites, the loss of which could result in the loss of many species.	
(B) Modularity
	Some species interact more frequently with some habitat patches, creating modules or compartments.	A modular structure (high modularity value) implies a high level of specialization of species for some habitats, and each habitat should be considered as a separate management unit. A non-modular structure (low modularity value), typical of random networks, implies that the protected area should be managed as a whole.	
Node level: habitat patch role	
(C) Patch centrality
	Central habitat patches are those that share many species with other habitat patches (patch A).	Central habitat patches (high centrality values) play a fundamental role in supporting generalist species across the whole species–habitat network. Non-central patches (low centrality values), however, could host unique sets of species.	

To exemplify the application of species–habitat networks to inform landscape management, we present a case study on the conservation of butterflies across a heterogeneous protected area in North-East Italy. We selected butterflies as model organisms as they are excellent indicators of habitat quality, being particularly sensitive to environmental changes (Thomas et al., 2004; WallisDeVries & Ens, 2010). Moreover, butterfly species largely vary in their life history traits such as mobility and phagy (Dennis, Shreeve & Van Dyck, 2003). The chosen protected area is composed of five major habitat types intermixed across the area, that is, three successional stages of dry calcareous grasslands along a natural disturbance gradient, that are usually the focus of conservation plans, and two managed grasslands, hay meadows and wet meadows, that can be seen as potential surrogate habitats to support butterfly diversity. We first applied traditional diversity analyses and then focused on unipartite and bipartite species–habitat network analyses. The aims of this study were: (i) to describe the emerging properties of the species–habitat network at the scale of the whole protected area, and (ii) to identify the key habitat patches for butterfly conservation across the protected areas, that is, those supporting the highest number of species and those with unique species assemblages (e.g., hosting specialist species). The information derived will help to tailor management plan for the protected area.

Materials and Methods

Study area

The study was carried out in the Friuli-Venezia Giulia region (North-East Italy), in the Special Protection Area “Magredi di Pordenone” (SPA-IT 33110011) (46°04′12.5″ N 12°45′46.5″ E). The area is part of the European Union Natura 2000 network, the largest coordinated system of protected areas in the world, covering over 18% of the EU land area. The size of the chosen protected area is c. 101 km2, and it includes four Sites of Community Importance: “Magredi di Tauriano” (SIC-IT3310008), “Magredi del Cellina” (SIC-IT 3310009), “Torbiera di Sequals” (SIC-IT 3310005) and “Risorgive del Vinchiaruzzo” (SIC-IT3310010). The bedrock consists of coarse alluvial calcareous-dolomitic sediments. The area is protected by the Natura 2000 network for its high value ecosystems (LIFE10 NAT/IT/000243), and it is characterized by a remarkable diversity of alluvial grassland habitats. We identified five main habitat types: three successional stages of dry semi-natural grasslands on calcareous substrate along a disturbance gradient, that is, (i) recently disturbed grasslands, with a low herbaceous cover (bare ground cover > 75%) and mainly composed by pioneer species, (ii) continuous grasslands, with intermediate natural disturbance, and a moderate herbaceous cover (10% < bare ground cover < 30%), and (iii) evolved grasslands, undisturbed for long time, and with a continuous herbaceous cover (bare ground cover < 10%) and presence of isolated shrubs; and two managed grasslands, that is, (iv) hay meadows, un-improved grassland mown twice a year and (v) wet meadows, mown once every 1–2 years (Table S1). The natural disturbance in dry calcareous grasslands is related to periodic floods that destroy the vegetation and the organic layer of the soil, halting the shrub encroachment. The continuous and evolved semi-natural dry grasslands are classified as Natura 2000 habitat 62A0 (Eastern sub-Mediterranean dry grasslands, Scorzoneretalia villosae).

Sampling design and butterfly sampling

We selected 44 sites, each belonging to one habitat type (Fig. S1). The number of sites for each habitat type was proportional to their cover in the protected area. We therefore selected 10 patches for each successional grassland stage and seven patches for both hay meadows and wet meadows. Each site covered an area of 2,500 m2 (50 × 50 m).

Adult butterflies (Papilionoidea) were surveyed five times between March and September 2010. Sampling occurred between 09.00 and 17.00 in days with favorable weather conditions (cloudiness < 25%, low or absent wind, air temperature > 18 °C). Each site was sampled for 15 min for each round. Surveys were always carried out by the same two operators, LM and Paolo Paolucci (University of Padua), which recorded all butterflies in the sampling area by visual sighting. Individuals that could not be identified while in flight were caught, identified and released at the end of the sampling. In each round, the order in which sites were sampled was randomized to avoid bias related to the time of sampling. Butterfly nomenclature follows Karsholt & Nieukerken (2011).

Data analyses

Diversity analyses

For each habitat patch, we calculated butterfly richness (total number of species) and evenness (Evar index) and used linear models to evaluate the effect of the habitat type on diversity indices. Both indices were calculated using the vegan package (Oksanen et al., 2019). All analyses were performed using R version 3.6.1 (R Development Core Team, 2019).

Species–habitat network analyses: bipartite network

We built a bipartite weighted network with patches and butterfly species as nodes, and calculated four network-level metrics providing complementary and non-redundant information: modularity, weighted NODF, robustness and connectance. Modularity describes how interactions between butterflies and patches are partitioned into separate modules, ranging between 0 (random network) and 1 (complete compartmentalized network) (Newman, 2006). Weighted NODF, the weighted Nestedness metric based on Overlap and Decreasing Fill, is the property by which specialist species interact with a subset of the sites that generalist species interact with, ranging between 0 (non-nested network) and 100 (perfectly nested network) (Almeida-Neto & Ulrich, 2011). We then checked for both metric significance using z-scores, calculated using 1,000 null models obtained with the Patefield algorithm (Dormann & Strauss, 2014). The two metrics provide fundamental information about network architecture (Bascompte et al., 2003; Olesen et al., 2007; Bastolla et al., 2009; Thébault & Fontaine, 2010; Tylianakis et al., 2010; Carstensen, Sabatino & Morellato, 2016; Grilli, Rogers & Allesina, 2016). Robustness, a measure of network stability against species extinction, was calculated removing butterfly species from the network from the rarest to the most abundant, ranging between 0 (highly unstable network) and 1 (highly stable network) (Memmott, Waser & Price, 2004). Connectance is a measure of network complexity which specifies the realized proportion of all possible links in a network, ranging between 0 (simple network) and 1 (complex network) (Dunne, Williams & Martinez, 2002). To compute network-level metrics, we used the bipartite package (Dormann, Gruber & Fründ, 2008).

Species–habitat network analyses: unipartite network

Starting from the bipartite species–habitat network, we built a unipartite weighted network, with patches as nodes and shared butterfly species as edges, that is, links between nodes. The weight of these links reflects the number of shared butterfly species between sites. Unipartite networks provide complementary information on network topology and node role in the network by using centrality metrics and community detection techniques developed for studying social networks (Freeman, 1978; Borgatti, 2015; Bedi & Sharma, 2016).

For each patch, we calculated weighted degree centrality, an index which specifies the role played by each patch within the network, highlighting the focal ones. It is based on both the number of connections with other patches and the average weight of these connections, adjusted by an α parameter (Opsahl, Agneessens & Skvoretz, 2010). We set the α parameter to 0.5, so patches with a higher number of connections have a stronger weighted degree centrality value (Rodríguez-Rodríguez, Jordano & Valido, 2017). A high centrality value indicates a patch which hosts many generalist species, while a low centrality value indicates a patch which hosts specialist or few species. We then used linear models to test the effect of habitat type on patch weighted degree centrality.

Moreover, to further investigate the structure of butterfly communities, we applied several community detection techniques. Community detection analysis is similar to modularity analysis in a bipartite network, but it is based on unipartite networks, so the result is a clusterization of patches based on the butterfly species they share. Because of the small network size (44 sites × 74 butterfly species) and the high value of the mixing parameter µ calculated using the multimodel algorithm (µ = 0.58), we selected two more algorithms for detecting communities, the spinglass algorithm and the walktrap algorithm (Yang, Algesheimer & Tessone, 2016).

We used the igraph package (Csardi & Nepusz, 2006) for building the unipartite weighted network and for community detection analysis, while weighted degree centrality was calculated using the tnet package (Opsahl, 2009).

Results

In the 44 sites, we sampled 6,273 adult butterflies belonging to 74 species and five families (Table S2). The most abundant species were Coenonympha pamphilus (1,022 individuals), Melanargia galathea (711 individuals) and Coenonympha arcania (491 individuals), while the most frequent ones were Pieris rapae (found in 32 sites), Coenonympha pamphilus (found in 31 sites) and Polyommatus icarus (found in 28 sites) (Table S2). We sampled two species included in the Habitats Directive annexes II and IV, Coenonympha oedippus (17 individuals in one site) and Lycaena dispar (11 individuals in one site), one species that is categorized as vulnerable in the Italian Red List for butterflies, Phengaris alcon (one individual) (Bonelli et al., 2018) and one species that is protected in the Friuli Venezia-Giulia region, Thecla betulae (two individuals in one site) (Valenti & Renzi, 2016) (Table S2). In each site, we found an average of 143 individuals (min = 2, max = 435) and an average of 17 butterfly species (min = 1, max = 32) (Table S3). The poorest habitat in terms of both butterfly abundance and richness was the disturbed grassland, with a total of 68 individuals belonging to 10 species. The richest one was the evolved grassland, with a total of 2,655 individuals belonging to 54 species (Table S3).

Whole network

The species–habitat network was complex, with highly connected habitat patches and butterfly species (connectance = 0.28), even if its size was relatively small (44 habitat patches × 74 butterfly species) (Fig. 1), so network structure was highly stable (robustness = 0.96). The network was significantly more modular than expected by chance (modularity = 0.35, modularity z-score = 95), and clusters coarsely matched habitat types, at least for the managed ones (Fig. S2). The modularity value, however, indicated a weak modular structure. On the other hand, the network was less nested than expected from the null models (weighted NODF = 25.04, weighted NODF z-score = -28.8). Community detection analysis confirmed the weighted NODF and modularity results. Both the multilevel and spinglass algorithms identified three communities (Figs. 2A and 2B), while the walktrap algorithm identified four communities (Fig. 2C). In general, the results of the three community detection algorithms converged and identified similar clusters. We can recognize three major communities: one for disturbed dry calcareous grasslands, one for un-managed grasslands (continuous and evolved dry calcareous grasslands) and one for managed grasslands (hay and wet meadows).

Figure 1 The bipartite species–habitat network.

Coloured nodes represent habitat patches, with patch code within the node (see Table S1). Black nodes represent butterfly species, with node size reflecting the number of links for each species. Grey links indicate species occurrence.

Figure 2 Community detection clusterization.

Community detection clusterization with (A) multimodel algorithm, (B) spinglass algorithm and (C) walktrap algorithm. The different colours indicate the communities detected by the different algorithms based on the shared species, while the numbers represent the a priori habitat classification based on the vegetation physiognomy.

Habitat level

Species richness, species evenness Evar, and patch weighted degree centrality were strongly related to habitat type (Figs. 3A–3C; Table 2). Disturbed grassland was the habitat with the lower species richness and centrality values, and the higher evenness. The number of butterfly species and the patch centrality values strongly increased along the grassland successional gradient, while evenness exhibited an opposite pattern. All three indices were comparable for evolved grassland and hay meadow, while only species evenness was similar for evolved grassland and continuous grassland.

Figure 3 Boxplots showing the effect of habitat type on (A) species richness, (B) species evenness Evar and (C) patch weighted degree centrality.

The lowercase letters a, b, c indicate significant differences from Tukey’s HSD test at P < 0.05.

Table 2 Results of the linear models testing the effect of habitat type on (A) butterfly species richness, (B) butterfly species evenness Evar and (C) patch weighted degree centrality.

		Estimate	SE	t-value	p-Value	
(A) Species
richness	Intercept (evolved grassland)	23.60	1.64	14.43	<0.01	
Hay meadow	1.54	2.55	0.61	0.55	
Continuous grassland	−5.80	2.31	−2.51	0.02	
Disturbed grassland	−20.30	2.31	−8.78	<0.01	
Wet meadow	−6.17	2.55	−2.42	0.02	
(B) Species evenness Evar	Intercept (evolved grassland)	0.37	0.04	8.81	<0.01	
Hay meadow	0.07	0.07	1.01	0.32	
Continuous grassland	0.08	0.06	1-34	0.19	
Disturbed grassland	0.49	0.06	8.19	<0.01	
Wet meadow	0.19	0.07	2.88	<0.01	
(C) Patch weighted
degree centrality	Intercept (evolved grassland)	128.60	5.60	22.95	<0.01	
Hay meadow	−1.66	8.73	−0.19	0.85	
Continuous grassland	−14.17	7.93	−1.79	0.08	
Disturbed grassland	−78.27	7.93	−9.88	<0.01	
Wet meadow	−31.21	8.73	−3.57	<0.01	

Patch level

Weighted degree centrality for patches was moderately high, with a mean value of 102.38 (min = 25, max = 150.39), because of the high number of connections between habitat patches. The ranking of patches based on their centrality values showed that the most central patches did not belong to a single habitat (Figs. 4A and 4B). In fact, the ten most central patches belonged to all habitat types except for disturbed grassland: four hay meadow patches, three evolved grassland patches, two continuous grassland patches, and one wet meadow patch. All disturbed grassland patches were peripherals. Species richness and evenness were strongly correlated to weighted degree centrality (Pearson’s correlation for patch centrality and species richness = 0.95, p-value < 0.01; Pearson’s correlation for patch centrality and species evenness = −0.87, p-value < 0.01), so the most central patches hosted more species and their abundance distribution was more uneven.

Figure 4 Weighted degree centrality.

(A) Patch ranking based on weighted degree centrality, and (B) map of the 44 sampling sites, with point size reflecting weighted degree centrality and indication of patch code (see Table S1). Map credit: © OpenStreetMap contributors.

Discussion

Here, we proposed to adapt ecological network tools to describe complex spatial interactions between species and habitats (Marini et al., 2019) and to use the resulting network metrics to improve conservation of butterfly species across a heterogeneous protected area. Despite the small size of the protected area, we found a remarkable diversity of butterflies, with 74 species, more than 25% of the total butterfly richness of Italy (Bonelli et al., 2018). The species–habitat network highlighted a general relaxed specialization of butterflies for habitats, indicating that species were affected by the management of the whole protected areas, beyond the boundaries of their preferred habitat type. The species–habitat network approach helped identifying both central habitat patches that were able to support the highest number of species and also habitat modules that supported rare specialist species.

Whole network

Network-level metrics can help to unveil the emergent properties of species–habitat networks. Modularity in bipartite networks plays an important role in network function, often improving community stability (Olesen et al., 2007; Tscharntke et al., 2007; Tylianakis et al., 2010; Grilli, Rogers & Allesina, 2016). In species–habitat networks, modules are composed of groups of tightly interacting species and patches. In our network, modularity was higher than expected by chance, and modules coarsely matched major habitat types. However, modularity was generally weak, indicating that several modules were still highly connected to each other. In particular, some habitats—the continuous and evolved grasslands – were visited by many species, and those species were mainly generalists. On the other hand, in our modularity analysis based on bipartite networks, four out of seven patches of wet meadow created a single, strong module due to the presence of specialist species such as Coenonympha oedippus and Lycaena dispar (Skórka, Settele & Woyciechowski, 2007). The removal of the wetland patches can therefore strongly affect the butterfly species pool of the whole protected area, being harmful for the persistence of rare, specialist species. The whole network, overall, was highly stable in terms of robustness to species extinction, as even specialist species were hosted in several habitat patches.

Differences in species assemblages between habitat types were confirmed by the community detection analysis. All detection algorithms yielded similar results and patches belonging to the same habitat almost always clustered together. The first community was roughly composed of only disturbed grassland patches, the second one was composed of calcareous dry habitat patches (continuous and evolved grassland) and the third one was composed of managed habitat patches (hay and wet meadows). It is therefore important to notice that community detection analysis, as all techniques that rely on unipartite networks, is exclusively based on shared species, and does not take into account the unshared ones, while modularity based on bipartite networks can identify key habitat patches for specialist species. For conservation purposes, it is therefore fundamental to apply both approaches to capture different facets of network organization. While modularity allowed to identify groups of patches where specialists are concentrated, centrality helped to identify the habitat patches that supported a larger number of generalists. Depending on the conservation aims, actions could focus on specific habitat patches or, on the contrary, manage the protected area as a whole, considering all habitat patches together.

Habitat level

Species richness, evenness and patch centrality differed among habitats. Disturbed grasslands had the lowest species richness and patch centrality, and the highest species evenness. The low herbaceous cover, low diversity of plant species and low flower availability of disturbed grasslands led to species-poor communities with even abundance distribution. The high evenness in disturbed grasslands was probably driven by the immigration of mobile and generalist species and by the low contribution to density from local recruitment (Marini et al., 2014). As the succession of grassland ecosystems proceeded, plant cover, plant richness and therefore butterfly species richness and patch centrality increased, with a consequent decrease in species evenness. Evolved grasslands were the most central habitat due to their considerable diversity of plant species and complex vegetation structure including both herbaceous species and shrubs (WallisDeVries, Poschlod & Willems, 2002; Ernst, Tscharntke & Batáry, 2017). Hay meadows, despite being impacted by mowing, hosted many species and were as central as evolved grasslands. The positive impact of low-intensity management on plant and butterfly communities has already been investigated (WallisDeVries & Raemakers, 2001; Silva et al., 2019), and a mosaic of managed and un-managed patches seems to be the best solution for maintaining biodiversity and network robustness. In fact, managed meadows are located in sites where floods, quite common in the study area, do not occur, safeguarding habitat patches suitable for a large number of butterfly species. The absence of flood disturbance is therefore a key driver of butterfly species diversity in hay meadows, despite the local disturbance of mowing. The central role of managed meadows also suggests that this habitat can contribute to increase the area of suitable habitat for the large majority of butterfly species considered typical of dry calcareous grasslands.

Patch level

Planning of conservation actions in protected areas often requires information about the role of single sites in supporting the focal biodiversity groups. The use of centrality measures to rank the importance of single patches has been extensively studied (Estrada & Bodin, 2008; Gilarranz et al., 2015; Poodat et al., 2015; Pereira, Saura & Jordán, 2017), as central nodes are known to promote stability in habitat networks (Thompson, Rayfield & Gonzalez, 2017). As explained above, evolved grasslands and hay meadows turned out to be fundamental habitats for butterfly conservation, but the ranking of individual patches based on weighted degree centrality also showed that central patches did not exclusively belong to these habitats. Furthermore, even within the same habitat, not all patches were equally relevant. This indicates that some patches can play an important role in the protected area irrespective of the habitat type. The most peripheral nodes were represented by both disturbed grassland and wet meadow patches, but while disturbed grasslands were always characterized by species-poor communities, wet meadows were rich in specialist species that were not shared with other habitats. As evolved grassland and hay meadow patches had a similar role in supporting butterfly communities within the protected area, several managed meadow patches can be seen as a surrogate habitat for dry semi-natural grasslands in supporting a large number of shared species. Centrality analysis can therefore be a useful tool to highlight the focal patches within a heterogeneous landscape and so to improve conservation planning.

Study limitations

There are two main limitations of this study. First, even if most butterflies were counted while foraging on flowers, some individuals were possibly using habitat patches only as stepping stones for dispersal. Traditional butterfly transect counts should therefore be complemented with more detailed information on how individuals interact with local resources. Second, our network analyses on modularity and community detection are not spatially explicit. Besides the habitat similarity effects, other drivers can contribute to form habitat modules across species-habitat networks. In particular, spatial autocorrelation can be an important process as habitat distribution across real landscapes is often non-random making difficult to disentangle pure habitat from pure spatial effects.

Conclusions

Developing conservation plans for protected areas across heterogeneous landscapes can be difficult. Here, we highlight the importance of an integrative approach, combining traditional diversity analysis and network analysis, for the identification of focal habitats and patches in a protected area. The species–habitat network of the protected area appeared to have a weak modular structure where the main habitat types tended to host different species assemblages. However, the habitat modules also shared a large proportion of species that are able to move and use resources across the whole protected area. Even butterfly species typically considered as habitat specialists according to the literature were actually observed across several habitats, suggesting that protecting them only within their focal habitat can be limiting. Calcareous dry grasslands are well-known key habitats for butterfly conservation (Silva et al., 2019), but we also pointed out the central role of agriculturally managed meadows across the protected area. Hay meadows, in particular, can act as a surrogate habitat for evolved calcareous grasslands patches, hosting surprisingly similar species assemblages. Although hay meadows are not currently considered as habitats with high conservation priority, more attention should be placed on the maintenance of their extensive management. Wet meadows emerged as the only habitats characterized by a distinctive module of wetland specialists. In conclusion, the protected area needs to be considered as a single dynamic unit to plan conservation actions.

Supplemental Information

Supplemental Information 1 Site map.

Map of the 44 sampling sites within the Special Protection Area “Magredi di Pordenone” (SPA-IT 33110011). Map credit: ©OpenStreetMap contributors.

Click here for additional data file.

Supplemental Information 2 Plot of modules for the bipartite species–habitat network.

Click here for additional data file.

Supplemental Information 3 Habitat information.

Number of sampling patches, management strategy, level of disturbance, bare ground cover, and shrub cover for each habitat type. * indicates habitat types included in the list of Natura 2000 protected habitats.

Click here for additional data file.

Supplemental Information 4 Sampled butterfly information.

List of sampled butterfly families and species with total abundance and frequency in the 44 sampling sites. * indicates protected and vulnerable butterfly species.

Click here for additional data file.

Supplemental Information 5 Abundance and richness of butterflies for each habitat type and habitat patch.

Click here for additional data file.

Supplemental Information 6 Species–habitat network raw data.

The species–habitat network, with butterfly species on columns and sites with the indication of habitat type on rows.

Click here for additional data file.

We thank Paolo Paolucci, Alessandro Fracasso and Luvi Dal Canton for their help during the field work.

Additional Information and Declarations

Competing Interests

Author Contributions

Data Availability

The authors declare that they have no competing interests.

Andree Cappellari analyzed the data, prepared figures and/or tables, authored or reviewed drafts of the paper, and approved the final draft.

Lorenzo Marini conceived and designed the experiments, performed the experiments, analyzed the data, prepared figures and/or tables, authored or reviewed drafts of the paper, and approved the final draft.

The following information was supplied regarding data availability:

The species–habitat network data is available in the Supplemental Information.

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
