# Peer review of "Improving insect conservation across heterogeneous landscapes using species–habitat networks"

_PeerJ, doi:10.7717/peerj.10563_

## Round 0.1 · original submission · Minor Revisions

I have now received reviews by two experts in the field who were very positive about the manuscript, an assessment with which I concur. Their comments should help mop up minor issues, clarify methods, and strengthen the discussion.

·

Basic reporting

The paper is very well written, clear to follow, and very interesting. There are a few minor typographic errors or missing words, so I suggest the authors check the manuscript thoroughly before publication.

I have one comment: as described in the methods, the study site is one protected area that contains four Natura 2000 sites. For most of the study, the authors refer to the one protected area as the overall habitat. I'm not 100% familiar with the Natura 2000 system, so I did find this a little confusing - for example, in Australia there are different types of protected areas, but they exist as independent areas..we don't generally have one type embedded within another type. Perhaps this could be clarified for readers not familiar with the system.

Experimental design

The study is an interesting application of an emerging method for understanding more complex relationships between species and their habitats. I am familiar with this method and found the experimental design here rigorous and well defined.

As someone who is familiar with this method, I was interested about the choice to use unipartite analysis for some aspects (L147). Could the authors perhaps include more detail to explain the reasoning for this and why unipartite was considered more useful here than bipartite? I think this would be helpful for others working on this emerging method of research.

L149: were the edges/links 'number of shared butterfly species'? I'm not clear how species information was included in the edges information.

Table 1: I think it would also be helpful for readers to include in this table some detail on the scale and meaning of the metric to assist interpretation, e.g. what do high vs low levels of these metrics mean? etc.

Validity of the findings

I think the results are valid and robust based on the aims and methods used. The authors have interpreted conclusions thoughtfully.

Additional comments

Nice paper!

·

Basic reporting

Manuscript is very well written in clear and concise English. I have made a few suggestions to reduce ambiguity (in the general comments section below), but none of these require major changes.

Experimental design

Nothing to add here. The experimental design is robust and methodology is rigorous and appropriate to address the research questions posed.

Validity of the findings

Nothing to add here. Results are clear and robust. The discussion and conclusions are appropriate for the results.

Additional comments

I thoroughly enjoyed reviewing this manuscript by Cappellari and Marini – I found it to be excellently written, with robust experimental design and clear results. The manuscript is very polished, with some complex ideas and methodologies that are presented in a simple and clear manner. The manuscript structure is concise, and the analyses used are robust and elegant. I must emphasise that this is a very timely piece of work following Marini et al 2019 and provides an elegant solution to an often complex and difficult problem in conservation ecology. Overall, I do not have any major criticisms of this work. My comments below are minor, and mainly concern clarity in the writing. Also, because this work is likely to be applicable to a general audience, the authors may need to provide more frequent and thorough explanations of the network jargon.

Minor comments:
Title: might be confusing for those who aren’t familiar with “Natura 2000 protected areas”. Perhaps rephrase or remove?
Line 18: you need to explain what “Natura 2000 protected areas” are here.
Line 21: give examples of the traditional diversity measures that you used here.
Line 22: again, you need to explain what unipartite and bipartite networks are for your more general/lay audience.
Lines 23-24: “key habitats for butterfly conservation” is a little vague. What do you mean by “key habitats” and how did you define these? Are they habitats/patches with exceptionally high conversation value, for example?
Lines 29-31: just because species were observed in a particular habitat type doesn’t mean they were using resources in those habitats. Or were they? Can you explain this a little more?
Line 33: “…providing key implications for conservation” is too vague. You need to elaborate on what these implications for conservation are.
Line 39: could you provide examples of the most common habitat types here?
Lines 55-72: would it be worth discussing “Hackett et al. 2019. Reshaping our understanding of the roles of species in landscape-scale networks. Ecology Letters. 22,1367-1377”. This paper uses a different network approach to your study but essentially seeks to address the same problem in a somewhat similar way.
Line 75: can you provide a little more explanation as to why butterflies are an excellent indicator of habitat quality?
Line 77: which life history traits? Can you give examples?
Lines 84-85: as indicated in the abstract, what do you mean by “key habitats” and how did you define these? You just need to give a little more context.
Lines 125-126: it’s not clear why you selected these two diversity indexes, especially considering that there are so many indexes available. I’m not saying that you should have looked at more, but you need to give a strong justification for using these two.
Line 150: you need to give a more detailed explanation of what centrality is, especially for those who aren’t familiar.
Lines 168-181: species names need to be italicised throughout.
Lines 197-198: do you mean that these indices were different from each other among habitat types?
Lines 222-223: again, it’s not clear whether these butterflies were using resources within the habitats that they were observed in, or if they were simply passing through. Can you clarify/expand on this?
234: weak compared to what? You need to provide some more context here.
Lines 240-241: some discussion around network stability might be useful here too?
Line 242: Differences in what?
Lines 250 -253: this all makes sense, but then what would be the directive/recommendation to conservation decision makers? I think you need to go beyond this broad statement and provide an example of what you would recommend based on your findings (i.e., are you saying that we allocate resources to conserve a certain subset of patches or that we need to spread resources to conserve/manage all patches?).
lines 260-262: this sentence is a little confusing. Over what time scale was this “evolution”? Do you have a relevant citation(s)?
Line 263: central in terms of the network structure or in a spatial sense?
Lines 270-271: so, is the key driver here low level/frequency of disturbance? Maybe you could explore that concept a little more.
Lines 283-284: again, I wonder if butterflies in some of these patches were simply passing through (perhaps due to the patch’s location relative to other more important/resource-rich patches) rather than using resources within the patch they were observed in? Does spatial location of a particular patch relative to other patches (and their importance) play a role in determining the importance of that patch for butterfly species? I guess this becomes quite circular.
Lines 290-291: I absolutely agree but feel this statement is too vague and not really that helpful in a practical sense. Can you take the discussion in this paragraph one step further and explain how your findings could be applied by a conservation practitioner (i.e., by giving an explicit example)?
Line 296: needs to be written in present tense.
Line 303: I’m not sure that “ineffective” is the right word here. Protecting only one or a subset of habitats might be suboptimal, but not necessarily “ineffective”.
Figure 1: what do the codes within each coloured node represent? Each figure needs to be stand-alone.
Figure 3: I wonder if you could indicate on this graph which habitats were statistically different from each other, as determined by your linear models? Package emmeans will do this for you (using contrasts/comparisons).
Figure 4: what do the different colours represent? This need to be explained in your figure caption.

---

## Round 0.2 · accepted · Accept

Many thanks for this revision, I see the authors have made good use of the constructive suggestions they have received, further strengthening an already very well written and interesting paper.